# Effects of Motilin Receptor Agonists and Ghrelin in Human *motilin receptor* Transgenic Mice

**DOI:** 10.3390/ijms20071521

**Published:** 2019-03-27

**Authors:** Tomoe Kawamura, Bunzo Matsuura, Teruki Miyake, Masanori Abe, Yoshiou Ikeda, Yoichi Hiasa

**Affiliations:** 1Department of Gastroenterology and Metabology, Ehime University Graduate School of Medicine, Toon City 791-0295, Japan; tomoe831118@yahoo.co.jp (T.K.); trk_719@yahoo.co.jp (T.M.); masaben@m.ehime-u.ac.jp (M.A.); yikeda@m.ehime-u.ac.jp (Y.I.); hiasa@m.ehime-u.ac.jp (Y.H.); 2Department of Lifestyle-related Medicine and Endocrinology, Ehime University Graduate School of Medicine, Toon City 791-0295, Japan

**Keywords:** motilin, ghrelin, erythromycin, gastric emptying

## Abstract

Gastrointestinal motility is regulated by neural factors and humoral factors. Both motilin and ghrelin improve gastrointestinal motility, but many issues remain unclear. We prepared human *motilin receptor* transgenic (Tg) mice and performed experiments evaluating the effects of motilin, erythromycin (EM), and ghrelin. EM and ghrelin promoted gastric emptying (GE) when administered either peripherally or centrally to Tg mice. Atropine (a muscarinic receptor antagonist) counteracted GE induced by centrally administered EM, but not that induced by peripherally administered EM. The administration of EM in this model promoted the effect of mosapride (a selective serotonin 5-hydroxytryptamine 4 (5-HT4) receptor agonist), and improved loperamide (a μ-opioid receptor agonist)-induced gastroparesis. The level of acyl-ghrelin was significantly attenuated by EM administration. Thus, we have established an animal model appropriate for the evaluation of motilin receptor agonists. These data and the model are expected to facilitate the identification of novel compounds with clinical potential for relieving symptoms of dyspepsia and gastroparesis.

## 1. Introduction

Gastrointestinal motility is regulated by contraction and relaxation of the smooth muscle of the gastrointestinal tract and by neural and humoral factors such as peptide hormones. Although erythromycin (EM), one of the motilides, is commonly used as a gastrointestinal function-improving agent, many issues, including EM’s mechanism of action, remain unclear. Both motilin and ghrelin act as promoters of gastrointestinal motility, but rodents lack a motilin and motilin receptor (MLNR) [1], making it difficult to study the function (and mechanism of action) of motilides in rodent animal models or to determine a correlation between motilide and ghrelin activities. Ghrelin and motilin possess close structural similarity, both in their precursor peptides and their receptors. The human ghrelin and motilin receptors share 52% overall amino acid identity, including 86% identity in the seven-transmembrane region [2,3,4,5,6]. Matsuura et al. have reported the molecular binding mechanism of motilin and EM to MLNR, as well as that of ghrelin to the growth hormone secretagogues receptor (GHSR) [7,8,9,10,11,12,13]. Similarities also are suggested by ghrelin and motilin’s predominant distribution in the upper gastrointestinal tract and their release during hunger to influence gastrointestinal functions. Takeshita et al. reported that, in the human gastrointestinal tract, motilin receptors were observed in the muscle layer and myenteric plexus, and ghrelin receptors were observed in the mucosa, muscle layer, and myenteric plexus [14]. However, important differences exist between ghrelin and motilin. Firstly, ghrelin does not readily bind to the motilin receptor, and vice versa [15,16]. Secondly, the non-gastrointestinal expression of ghrelin and ghrelin receptors contrasts with very limited non-gastrointestinal expression of motilin and motilin receptors. Finally, the existence of different ghrelin receptor states, the possible existence of ghrelin receptor-like receptors, and the production of multiple relevant peptides (including ghrelin, desacyl-ghrelin and obestatin) from the *ghrelin* gene suggests that the ghrelin system has more extensive roles than does the motilin system [4,6,17]. The present study sought to clarify the central and peripheral mechanisms of action of motilides, and to identify a possible correlation between the motilin and ghrelin systems. Human *MLNR* transgenic (Tg) mice were constructed and studied. In the absence of ligands, the Tg mice were indistinguishable from isogenic wild-type (WT) animals; furthermore, administration of EM either centrally or peripherally did not change the gastric emptying (GE) responses in the WT mice. Mosapride citrate, a selective serotonin 5-hydroxytryptamine 4 (5-HT4) receptor agonist, also activates gastrointestinal tract motility; this drug is used clinically to treat gastrointestinal motility disorders. Loperamide is known to reduce gastrointestinal motility and is an effective treatment for diarrhea; this drug’s activity is mediated by the stimulation of μ-opioid receptors. We also investigated the interactions between EM and these agents.

## 2. Results

### 2.1. MLNR is Expressed Ubiquitously in Human MLNR Transgenic Mice

The human *MLNR* cDNA was incorporated into a transgene designed to direct ubiquitous expression in mouse. Mouse *GHSR* was expressed natively. *MLNR* and *GHSR* mRNA levels were evaluated by real-time reverse transcriptase PCR (RT-PCR) (Figure 1). *MLNR* mRNA was expressed primarily in the gastrointestinal tracts, but also was present ubiquitously in brain and liver. In contrast, *GHSR* mRNA was expressed primarily in the brain (mostly in the hypothalamus); low expression was observed in the gastrointestinal tract. We also examined the expression of *ghrelin* mRNA, which was most strongly expressed in the stomach (Figure 2) and is consistent with past reports [18]. Thus, the expression pattern of *MLNR* and *GHSR* in Tg mouse was similar to that in human.

### 2.2. EM and Ghrelin Promote Gastric Emptying when Administered Either Peripherally and Centrally in Tg Mice

We next tested the effects of EM administered by either the central (intracerebroventricular (icv); 3 µg EM in a 2 µL volume) or peripheral (intraperitoneal (ip); 10 mg/kg) route, and of ghrelin administered by the same routes (3 µg/2 µL and 0.5 mg/kg, respectively). The GE of a carboxymethyl cellulose/phenol red dye non-nutrient meal was significantly faster in mice administered peripherally with EM (0.51 ± 0.10 (mean ± SD), *n* = 6) or ghrelin (0.60 ± 0.08, *n* = 3) compared to that in animals dosed ip with vehicle (0.31 ± 0.13, *n* = 10) (*p* < 0.05) (Figure 3a). Similarly, GE was significantly faster in mice administered centrally with EM (0.44 ± 0.15, *n* = 14) or ghrelin (0.52 ± 0.11, *n* = 9) compared to animals dosed icv with vehicle (0.32 ± 0.10, *n* = 16) (*p* < 0.01) (Figure 3b). 

### 2.3. EM Promotes GE Both through the Vagus Nerve System and by Direct Action on MLNR

While GE was induced by centrally administered EM, this effect was attenuated by co-administration of atropine (Atr; 3 mg/kg). Specifically, central administration of EM only, EM + Atr, vehicle only, and vehicle + Atr yielded GE values of 0.47 ± 0.07, 0.19 ± 0.13, 0.33 ± 0.08, and 0.13 ± 0.11 (*n* ≥ 3), respectively (Figure 4a). In contrast, co-administration of Atr did not affect GE induced by peripherally administered EM, with peripheral administration of EM only, EM + Atr, vehicle only, and vehicle + Atr yielding GE values of 0.55 ± 0.10, 0.50 ± 0.22, 0.34 ± 0.14, and 0.18 ± 0.12 (*n* ≥ 7), respectively (Figure 4b). Previous work has shown that EM’s effect on the contractile response in the stomach of Tg mice reflects direct action via MLNR [19]. We therefore concluded that EM promotes GE both through the vagus nerve system and by direct action on MLNR. 

### 2.4. EM Potentiates the Effect of Mosapride

We investigated whether EM could also promote additional GE. EM (8 mg/kg) and mosapride (1 mg/kg) were co-administered by the ip route. Dosing of the mice with vehicle only, mosapride only, EM, and mosapride + EM yielded GE values of 0.32 ± 0.13, 0.52 ± 0.11, 0.48 ± 0.09, and 0.64 ± 0.09 (*n* ≥ 8), respectively (Figure 5). We therefore concluded that the effect of EM is additive with that of mosapride. 

### 2.5. EM Improves Gastroparesis Induced by Loperamide

We examined the potential utility of EM in enhancing gastroparesis induced by loperamide. EM (10 mg/kg) and loperamide (10 mg/kg) were co-administered by the ip route. Dosing of the mice with vehicle only, EM only, loperamide only, and loperamide + EM yielded GE values of 0.41 ± 0.13, 0.55 ± 0.10, 0.18 ± 0.16, and 0.58 ± 0.10 (*n* ≥ 6), respectively (Figure 6). We therefore concluded that EM improves loperamide-induced gastroparesis.

### 2.6. Motilides and Ghrelin May Play Complementary Roles

The levels of both acyl-ghrelin and motilides have been shown to correlate not only with appetite, but also with gastric motility [5,20]. We confirmed that, in the Tg mice, the plasma level of acyl-ghrelin during fasting is higher than that in the post-prandial state (Figure 7a).

The plasma level of acyl-ghrelin was investigated in Tg mice administered centrally with EM. Plasma samples were collected at 1 h after administration by the icv route. Acyl-ghrelin was detected at 17.65 ± 10.25 fmol/mL (*n* = 6) in fasting-group mice, and at 21.62 ± 0.92 fmol/mL (*n* = 3) and 5.95 ± 0.90 fmol/mL (*n* = 4) at 1 h after dosing with vehicle or EM, respectively (Figure 7b). Thus, acyl-ghrelin level was significantly (*p* < 0.05) attenuated in animals administered with EM by the central route. 

To confirm these results, the stomachs of the post-dose mice were evaluated by immunohistochemical (IHC) staining for ghrelin. Notably, significantly (*p* < 0.01) fewer ghrelin-positive cells were detected in the stomachs of the EM-administered mice (*n* = 5) than in those of the vehicle-administered mice (*n* = 5) (Figure 8). This result suggested that motilides and ghrelin may play complementary roles. 

## 3. Discussion

The gastrointestinal hormone motilin was identified over 40 years ago [21] following suggestions that a substance that regulated motility was released from the duodenum [22]. Motilin is a 22-amino-acid peptide, synthesized and secreted by endocrine cells in the epithelium of the human upper small intestine, most notably the jejunum and duodenum [23]. Motilin plays a role in initiating phase III of gastric migrating myoelectric complexes (MMC); on the other hand, ghrelin has roles in increasing appetite and also in gastric emptying [5,24,25].

There are some reports that motilin induces endogenous release of acetylcholine by activating the 5-hydroxytryptamine 3 receptors on vagal afferent nerves [26], and that motilin causes contractions by stimulating receptors on or in gastrointestinal muscle cells [27]. However, it is now understood that low concentrations of motilin receptor agonists increase myenteric cholinergic activity by facilitating acetylcholine release, whereas high concentrations of motilin receptor agonists directly induce muscle contraction. This cholinergic activity has been demonstrated in rabbit stomach [16,28] and in human stomach, where the most pronounced effects are seen in the human antrum [29]. Peripheral administration of the motilin receptor agonists can induce effects via any of 3 pathways, acting through afferent neurons, through efferent neurons, or by direct action on the muscles. In contrast, central administration of the motilin receptor agonists can induce effects only via one pathway, through efferent neurons. Our results support those of previous studies [19,26,27,28], which showed how motilide acts and indicated that centrally administered motilides do not infiltrate through the blood-brain-barrier. 

Although many reports have been published regarding the mechanisms by which motilin receptor agonists increase gastric cholinergic activity, these mechanisms have taken many years to resolve. Progress was impeded by the absence of a functional motilin system in laboratory rodents, as well as by the ongoing focus on the ability of motilin to induce directly the contraction of gastrointestinal smooth muscle. A small number of studies have claimed that active motilin receptors exist in rodents, suggesting that the activity of EM is mediated by binding to central motilin receptors that might be involved in the regulation of gastric motility in rodents [30,31,32,33]. However, we observed no GE response when EM was administered to wild-type mice (Appendix A), a result consistent with those of other studies that also failed to detect a functional motilin system in rodents [34,35].

To our knowledge, there have been no previous reports that EM has an additive effect with other compounds, nor that EM can improve gastroparesis. Thus, the demonstration here that EM and mosapride exhibit additive effects appears to be the first such demonstration. However, the results for this assay should be considered preliminary, given that the selected EM dose (8 mg/kg) was chosen based on an empirical judgment made under our experimental conditions. 

The work reported here began when we tried to establish a diabetic gastroparesis model, which would be characterized by delayed GE in the absence of obstruction. We were able to create a diabetic mouse model with the Tg mice by injecting animals with streptozotocin or by feeding the animals with a high-fat-diet; while some of the resulting mice showed delayed GE, others showed normal GE (Appendix A). These results are consistent with those of a previous report [36]. The variable symptomology is thought to reflect differences in the production of inflammatory factors by macrophages, leading to a loss of interstitial cells of Cajal [36]. In subsequent work, we tried to establish a gastroparesis model by treating the Tg mice with cisplatin (CDDP), but none of the resulting animals showed delayed GE. Therefore, we turned to the creation of a model of gastroparesis induced by loperamide. Loperamide acts on the same set of opioid receptors targeted by morphine. EM therefore showed efficacy in treating the symptoms of functional gastrointestinal disorders that do not improve with mosapride, as well as to show activity against opioid-induced bowel dysfunction (OBID).

There are a few reports examining the relationship between motilin and ghrelin [37,38]. In the present study, human *MLNR* Tg mice were administered EM by the central route and the plasma level of acyl-ghrelin was examined at 1 h post-dose. We observed a decrease in the plasma level of acyl-ghrelin, a result that was consistent with detection by IHC of a decreased number of ghrelin-positive cells in the stomachs of the EM-dosed mice (compared to those of vehicle-dosed animals). However, in mice dosed centrally with EM, we did not detect significant changes in the accumulation of *GHSR* mRNA in the hypothalamus, or in the GHSR protein level or the number of GHSR-positive cells (by IHC) in the brain cortex or stomach (Appendix A). We also did not detect, in mice dosed centrally or peripherally with ghrelin, significant changes in the accumulation of *MLNR* mRNA in the hypothalamus, stomach, or jejunum (Appendix A). The mechanism of EM’s effect on ghrelin production is unclear, although we hypothesize that the motilin and ghrelin systems have a compensatory relationship with each other. It will be important to elucidate the phenomenon whereby ghrelin production declines following administration of EM.

There are several limitations to the work presented here. Firstly, GE was evaluated by measuring the passage of phenol red. Several alternative methods have been used to evaluate gastric motility in rodents, including the ^13^C breath test [39], scintigraphy [40,41], and magnetic resonance imaging (MRI) [42]. We selected the conventional phenol red method based on its ease-of-use and low cost. Good correlation has been reported between the ^13^C breath test and the phenol red method [43]. One large advantage of testing GE using the ^13^C breath test would be its non-invasive nature, which would permit maintaining the mice under better conditions and repeated evaluation of GE in the same animals. Secondly, our current experiment was performed using only EM; motilin was not employed. The basis for this distinction between the effect of EM and motilin in our model is unknown, and contrasts with a report showing that EM and motilin have similar effects on isolated stomach [19]. These results suggest that, in vitro (ex vivo), EM and motilin behave similarly, presumably via direct action on MLNR, but that, in vivo, additional elements affect the response to these drugs. Thirdly, the experiments described here assessed changes to GE in response to acute administration; additional experiments addressing the effects of chronic administration will need to be performed.

## 4. Materials and Methods 

### 4.1. Animals

Ten-week-old male C57BL/6J (wild-type; CLEA Japan, Inc. Tokyo, Japan) mice and human *MLNR* Tg mice were housed individually on 12-h/12-h light/dark cycle with *ad libitum* access to water and a standard rodent diet (MF, Oriental Yeast Co., Ltd., Tokyo, Japan. Energy 359 kcal, water 7.9 g, protein 23.1 g, fat 5.1 g/100 g MF). Mice were deprived of food for 24 h before the start of each experiment. 

The Tg mice were generated using a full-length (1236-bp) cDNA of human *MLNR* [8]. This cDNA was inserted downstream of the *CAG* promoter of pCAGGS (Kaketsuken, Kumamoto, Japan), and the resulting plasmid was used for pronuclear injection into fertilized C57BL/6J oocytes as described previously [15]. Resulting progeny were tested using DNA isolated from mouse tails as a template in PCR screening for the presence of the human *MLNR* cDNA. Specifically, Tg pups were identified by PCR using a forward primer (5′-ATGGGCAGCCCCTGGAACGGC-3′) corresponding to the upstream end of the human cDNA and a reverse primer (5′-TCCCATCGTCTTCACGTTAGC-3′) corresponding to the downstream end of the human cDNA This primer pair was expected to yield a 1236-bp product if the human *MLNR* cDNA was present. 

All animal experimental protocols were approved by the Institutional Animal Care and Use Committee of the Ehime University Graduate School of Medicine (Permit Number: 05TI80-16, 1 March 2016).

### 4.2. RNA Isolation and Real-Time PCR 

RNA was isolated using RNeasy Plus Universal Kits (QIAGEN, Hulsterweg, Netherlands). RNAs were converted into cDNAs using a High-Capacity cDNA Reverse Transcription Kit (Thermo Fisher Scientific, Waltham, MA, USA). cDNA products were quantified by real-time PCR using Light Cycler SYBR Green (Roche, Basel, Switzerland) and a Light Cycler®480 (Roche, Basel, Switzerland), with the gene encoding glyceraldehydes-3-phosphate dehydrogenase (*GAPDH*) as an internal control. The forward and reverse mouse *GAPDH* primers were 5′-TGCACCACCAACTGCTTA-3′ and 5′-GGATGCAGGGATGATGTT-3′, respectively, which should yield a 177-bp product. The forward and reverse human *MLNR* primers were 5′-TCCTCTACAACCTCATTTCA-3′ and 5′-TTATCCCATCGTCTTCAC-3′, respectively, which should yield a 182-bp product. The forward and reverse mouse *GHSR* primers were 5′-GACCAGAACCACAAACAGACAG-3′ and 5′-GGCTCGAAAGACTTGGAAAA-3′, respectively, which should yield a 113-bp product. The forward and reverse mouse *ghrelin* primers were 5′-TCCTCTGGGAAGAGGTCAAA-3′ and 5′-CTGAGGCGGATGTGAGTTC -3′, respectively, which should yield a 103-bp product. 

### 4.3. Drugs and Chemicals

A liquid gastric load was prepared by formulating 50 mg phenol red in 100 mL of 1.5% carboxymethyl cellulose (CMC). Erythromycin (EM) and ghrelin were purchased from Mylan N.V. (Tokyo, Japan) and Wako Pure Chemical Co. (Osaka, Japan), respectively. Atropine was purchased from Tanabe Pharmaceutical (Osaka, Japan). Mosapride citrate and loperamide were purchased from Dainippon Sumitomo Pharma Co. (Osaka, Japan) and Wako Pure Chemical Co., respectively.

### 4.4. ELISA (Enzyme-Linked Immuno-Sorbent Assay)

Following collection, the blood samples were promptly centrifuged at 4 °C, and the supernatants were acidified with 0.1 Ma N HCl (1/10 volume). Plasma acyl-ghrelin levels were determined using the Sceti (Tokyo, Japan) ELISA kit for the detection of acyl-ghrelin.

### 4.5. Gastric Emptying (GE)

#### 4.5.1. Experimental Protocols

We assessed GE in 24-h-fasted mice according to the phenol red method, which has been described previously [44,45,46,47]. The animals (wild-type and human *MLNR* Tg mice) were administered 0.5 mL of test meal by oral gavage, and then were euthanized by cervical dislocation either immediately (t = 0) or 20 min after gavaging. Following laparotomy, the pylorus and the cardia were ligated, and the stomach was excised. Intracerebroventricular (icv) and intraperitoneal (ip) administration of drugs were performed 30 min before the gavage.

#### 4.5.2. Estimation of GE

The stomach was cut into pieces and homogenized, with its contents, in 25 mL of 0.1 N NaOH. The homogenate was centrifuged 2500× *g* for 3 min at 4 °C); an 8-mL aliquot of the resulting supernatant was combined with 1 mL of 33% trichloroacetic acid to precipitate proteins. After another round of centrifugation (2500× *g* for 30 min at 4 °C), 2 mL of 2 N NaOH were added to the supernatant, and the amount of phenol red was determined by measuring absorbance at 560 nm. This absorbance correlates with the concentration of phenol red in the stomach, which in turn depends on GE. The GE rate (in percent) was calculated as GE = (1−X/Y) × 100, where X and Y are the absorbances of phenol red recovered from the stomach of animals killed 20 min and 0 min (respectively) after gavaging. 

### 4.6. Administration of EM, Ghrelin, Mosapride, and Loperamide 

EM and ghrelin were administered by intracerebroventricular (icv) or intraperitoneal (ip) injection. The dose of EM was 3 µg/2 µL and 10 mg/kg, respectively and that of ghrelin was 3 µg/2 µL and 0.5 mg/kg, respectively. Mosapride (1 mg/kg) and loperamide (10 mg/kg) were administered only via the ip route.

For the icv injections, each mouse was first stereotaxically implanted with a single sterile guide cannula that was placed into the ventricle using the following coordinates for the tip of the cannula: 0.5 mm posterior and 1 mm lateral to the bregma, and 3 mm ventricle to the surface of the skull. Animals were subjected to inhalation anesthesia with isoflurane for the implantation surgery.

### 4.7. Histological Examination

Age-matched progeny (5 for each group) were killed by decapitation. The whole stomach was removed and opened through the greater curvature. Specimens intended for histological examination were fixed by immersion in 10% formaldehyde, processed, embedded in paraffin, sectioned, and stained with hematoxylin and eosin (H&E) using standard methodologies. For immunohistochemical staining of ghrelin, 4-µm sections prepared from paraffin-embedded blocks were pretreated at 125 °C in EDTA buffer (pH 9) for antigen retrieval. The sections then were incubated for 10 min with 10% normal goat serum (to block nonspecific binding) before incubation overnight at 4 °C with a 1:8000 dilution of a rabbit monoclonal antibody (ab 209790, Abcam Plc. Tokyo, Japan) raised against a peptide corresponding to the N-terminus (aa 1–100) of human ghrelin. Intrinsic peroxidase activity was blocked using a methanol solution containing 1% hydrogen peroxide, and the sections then were treated with MAX-PO (R) for 1 h. Finally, the sections were immersed in 3,3′-diaminobenzidine, and nuclear staining was performed with hematoxylin. All sections were reviewed by expert pathologists, and the number of ghrelin-positive cells/10 HPF (high power field) was assessed. 

### 4.8. Statistical Analysis

Values are expressed as mean ± SD. Statistical analyses were performed using the statistical software JMP® 9 (SAS Institute Inc., Cary, NC, USA). For analysis of three or more groups, analysis of variance (ANOVA) tests were performed with the Tukey post-hoc test, and analysis of differences between two normally distributed test groups was performed using Student’s *t*-test. A *p* value less than or equal to 0.05 was considered significant.

## 5. Conclusions

We have established an animal model appropriate for evaluation of motilin receptor agonists, providing an in vivo model that mimics gastric emptying in human. Our findings demonstrate the mechanism of improved gastric emptying induced by EM and possibly by motilides, and the relationship between motilin and ghrelin. These data are expected to facilitate the development of a novel drug that may ease the symptoms of dyspepsia and gastroparesis. 

## Figures and Tables

**Figure 1 ijms-20-01521-f001:**
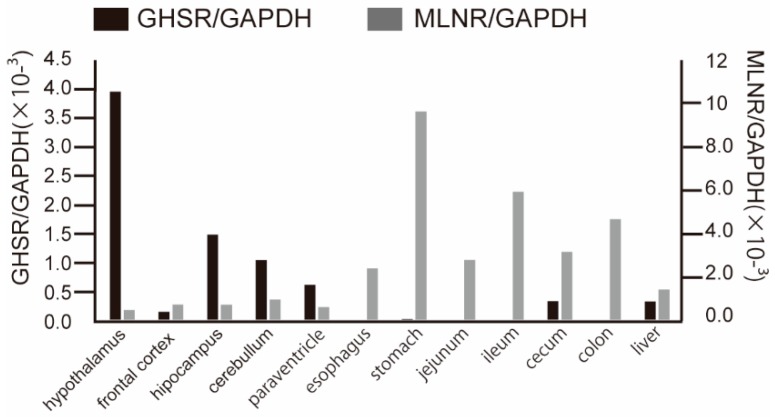
Accumulation of *MLNR* and *GHSR* transcripts in various tissues of human *MLNR* Tg mice (*n* = 3). The values shown are the numbers for real-time RT-PCR products of *MLNR* and *GHSR* mRNAs normalized to those of *GAPDH*.

**Figure 2 ijms-20-01521-f002:**
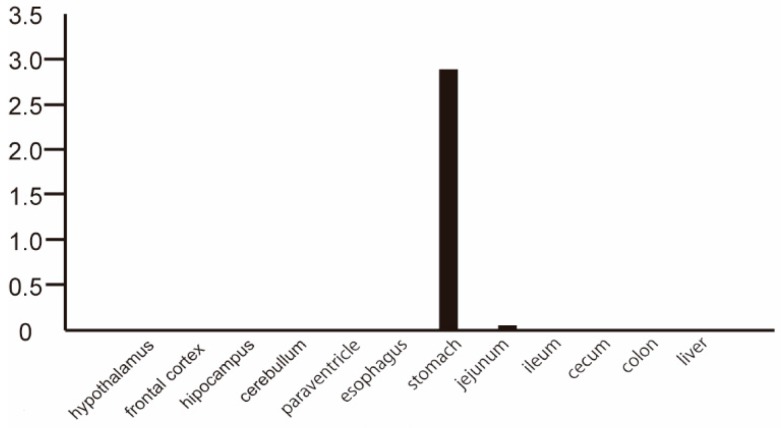
Accumulation of *ghrelin* transcripts in various tissues of human *MLNR* Tg mice (*n* = 3). The values shown are the numbers for real-time RT-PCR products of *ghrelin* mRNAs normalized to those of *GAPDH*.

**Figure 3 ijms-20-01521-f003:**
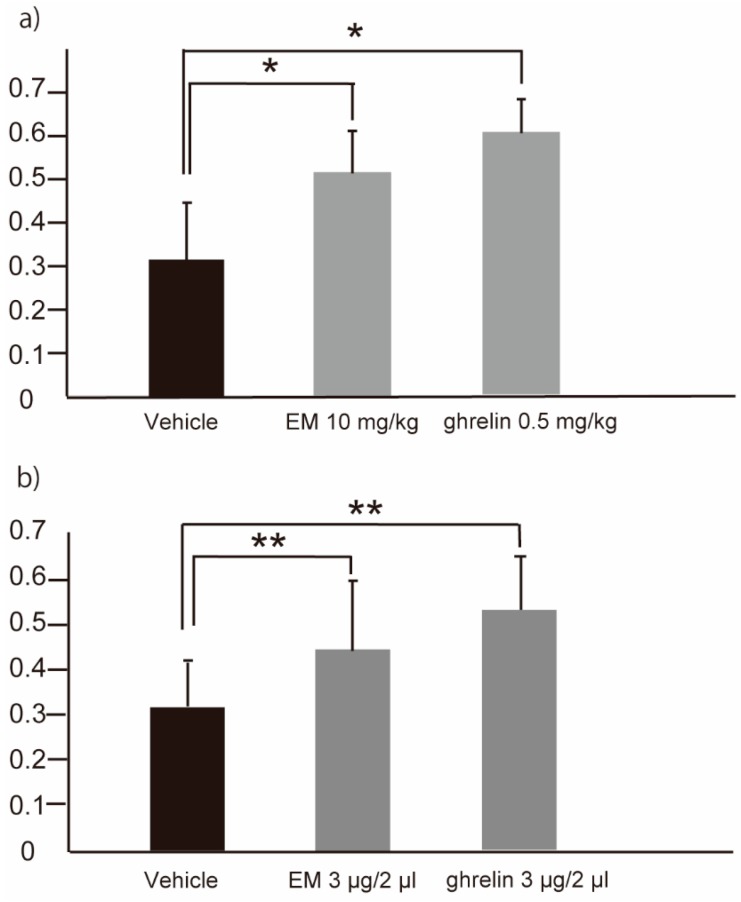
Gastric emptying of human *MLNR* Tg mice following peripheral (intraperitoneal) administration of EM (*n* = 6) or ghrelin (*n* = 3) (**a**), or following central (intracerebroventricular) administration of EM (*n* = 14) or ghrelin (*n* = 9) (**b**). Values represent the mean ± SD for the indicated number of animals. * *p* < 0.05, ** *p* < 0.01 (One-way ANOVA with Tukey post-hoc test).

**Figure 4 ijms-20-01521-f004:**
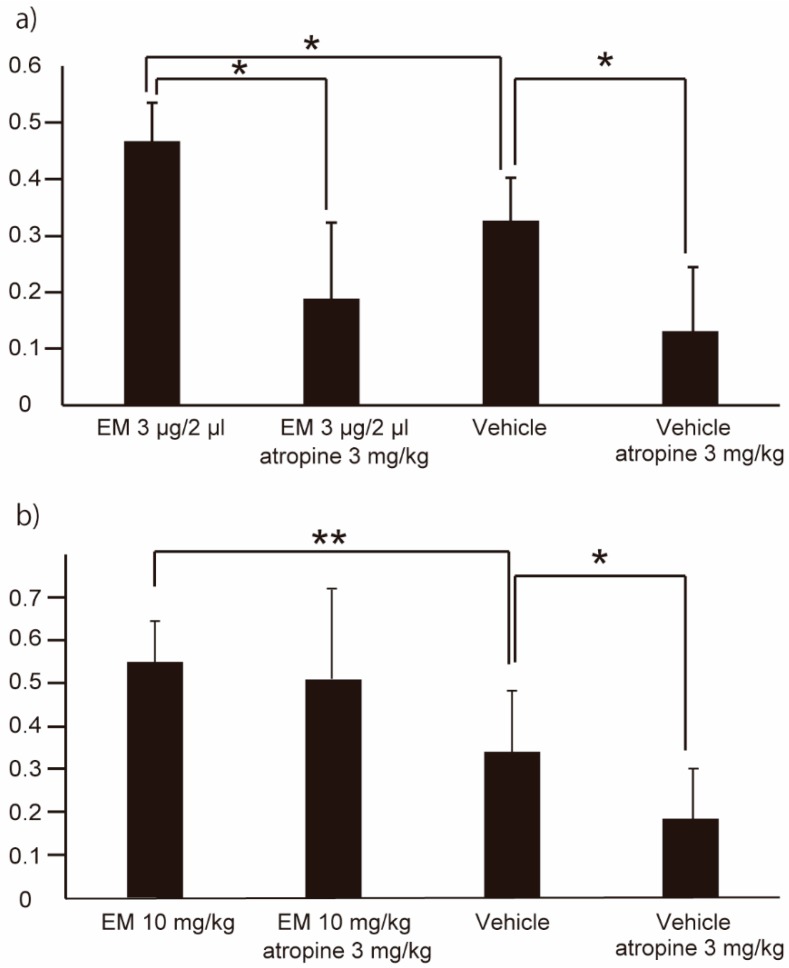
Gastric emptying of human *MLNR* Tg mice following central (intracerebroventricular) administration of EM only (*n* = 3), EM + atropine (*n* = 4), vehicle only (*n* = 6), or vehicle + atropine (*n* = 4) (**a**), or following peripheral (intraperitoneal) administration of EM only (*n* = 10) EM + atropine (*n* = 7), vehicle only (*n* = 14), or vehicle + atropine (*n* = 7) (**b**). Values represent the mean ± SD for the indicated number of animals. * *p* < 0.05, ** *p* < 0.01 (One-way ANOVA with Tukey post-hoc test).

**Figure 5 ijms-20-01521-f005:**
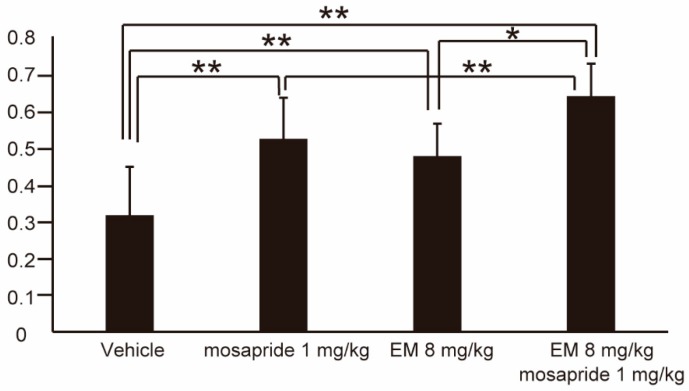
Gastric emptying of human *MLNR* Tg mice following peripheral (intraperitoneal) administration of vehicle only (*n* = 10), mosapride (*n* = 9), EM only (*n* = 8), or EM + mosapride (*n* = 10). Values represent the mean ± SD for the indicated number of animals. * *p* < 0.05, ** *p* < 0.01 (One-way ANOVA with Tukey post-hoc test).

**Figure 6 ijms-20-01521-f006:**
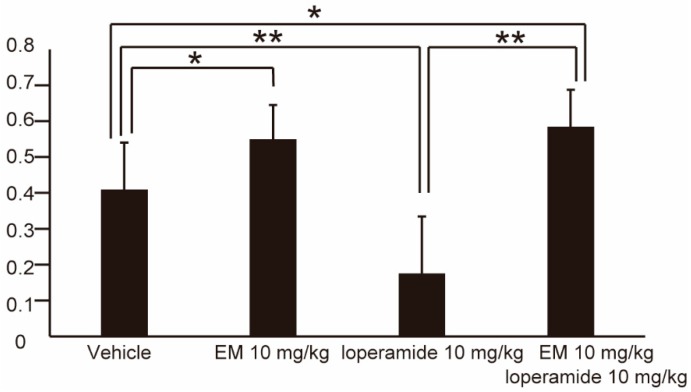
Gastric emptying of human *MLNR* Tg mice following peripheral (intraperitoneal) administration of vehicle only (*n* = 14), EM only (*n* = 10), loperamide only (*n* = 8), or EM + loperamide (*n* = 6). Values represent the mean ± SD for the indicated number of animals. * *p* < 0.05, ** *p* < 0.01 (One-way ANOVA with Tukey post-hoc test).

**Figure 7 ijms-20-01521-f007:**
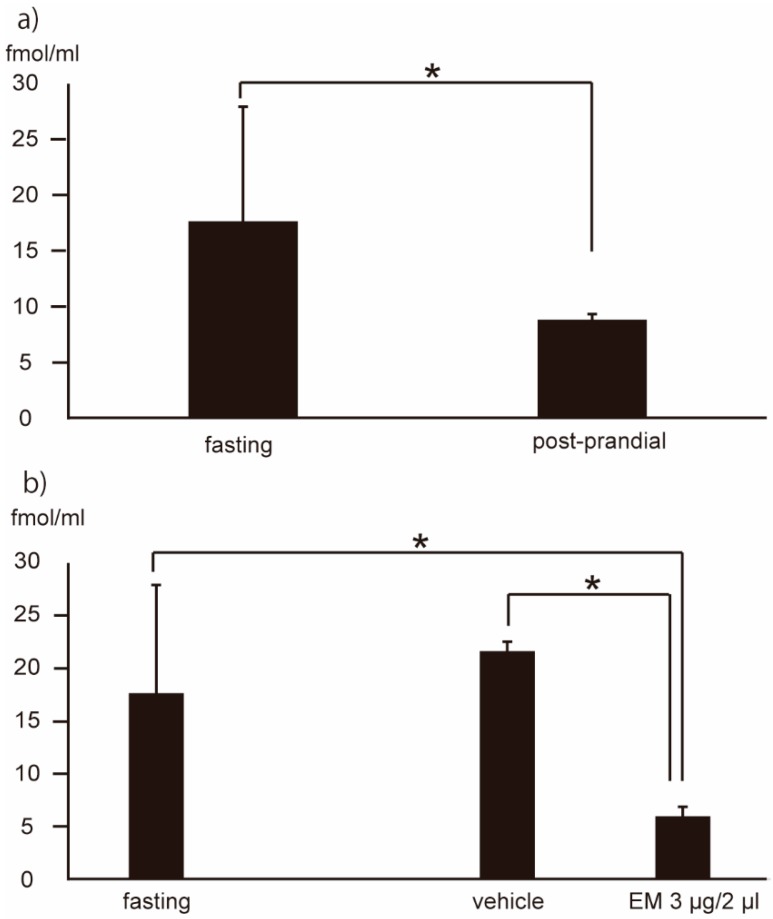
Plasma levels of acyl-ghrelin in fasting and post-prandial state (**a**), and 1 h following central (intracerebroventricular) administration of EM (*n* = 4) or vehicle (*n* = 3) (**b**). Values represent the mean ± SD for the indicated number of animals. * *p* < 0.05 (Unpaired Student’s *t*-test, One-way ANOVA with Tukey post-hoc test).

**Figure 8 ijms-20-01521-f008:**
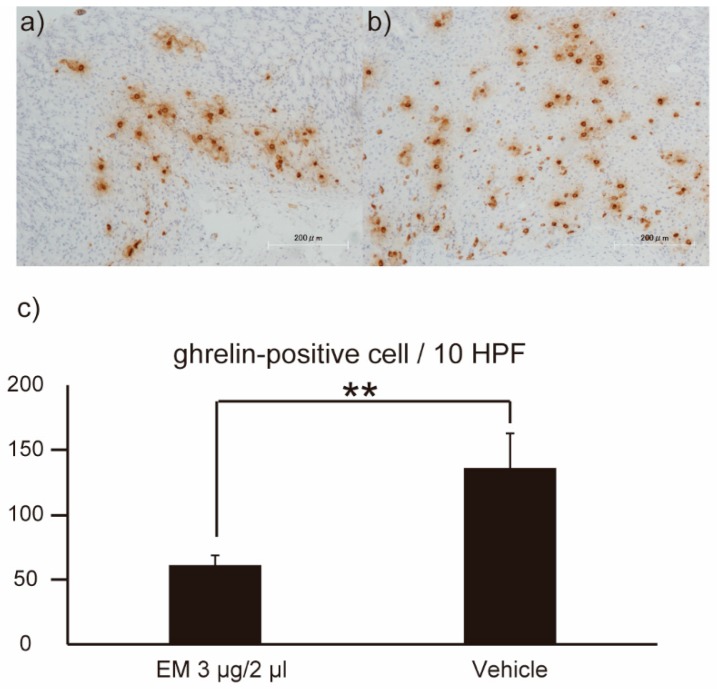
Immunohistochemical staining for ghrelin expression in the stomachs of mice. Representative micrographs are provided showing the distribution of ghrelin-positive cells (brown staining) in the stomachs of mice administered EM (**a**) or vehicle (**b**). The numbers of ghrelin-positive cells/10 high power field (HPF) were quantified in stomachs from *n* = 5 mice (**c**). Values represent the mean ± SD for the indicated number of animals. ** *p* < 0.01 (Unpaired Student’s *t*-test).

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
