# Peer review of "Effects of Motilin Receptor Agonists and Ghrelin in Human motilin receptor Transgenic Mice"

_ijms, 2019, doi:10.3390/ijms20071521_

Round 1
Reviewer 1 Report
The authors aimed to determine the effects of erythromycin and ghrelin on gastric emptying in human motilin receptor Tg mice, to establish an animal model appropriate for the evaluation of motilin receptor agonists. The authors found that EM and ghrelin affect GE in Tg mice, and EM was able to enhance the effect of mosapride, and counteract loperamide-induced gastroparesis. Furthermore, the authors speculate that there is a complementary effect between EM and ghrelin. The authors present their findings clearly. I have a few comments.
Major comment:
I have some issues with the statistical analyses of this study. In the study, multiple treatments (I.e. 3 or 4) are compared for the results presented in figure 3-6, however, the authors have performed pair-wise comparisons between all the treatments. To adjust for multiple comparisons, ANOVA should be performed, or, if the data are non-parametric, Kruskal-Wallis is the appropriate alternative. Only if a significant overall treatment effect is found in the analysis, the authors are allowed to perform comparison between all the individual treatments. Please refer to Festing et al, 2002, Guidelines for the design and statistical analysis of experiments using laboratory animals, ILAR journal, 43(4), 244-258. Also please describe the software and version used for analysis.
Minor comments:
1. In the abstract, please describe in an introductory sentence what atropine, mosapride and loperamide are.
2. In the result section, there are a few sentences that should be moved to the introduction, i.e. line 84-86 (mosapride citrate… promote additional GE), line 92-94 (loperamide is known… induced by loperamide), and line 100-101(the levels of…. With gastric motility).
3. Similarly, there are a few conclusive lines in the results section, which should be moved to the discussion, i.e. line 80-82 (previous work has….action on MLNR), line 89-90 (we therefore concluded…that of mosapride), line 97-98 (we therefore concluded… loperamide-induced gastroparesis), line 111-112 (this result suggested…play complimentary roles).
4. In the results, the authors talk about ‘higher’ gastric emptying. Would you consider changing ‘higher’ into ‘faster’, for clarity (line 68/70)?
5. The authors conclude in line 81/82 that EM promotes GE through the vagus nerve and by direct action in MLNR, which is in agreement with the results presented in figure 3. However, this statement does not explain why EM is only able to counteract the slowing effects of atropine on GE after ip administration, and not after icv admininstation. Please elaborate how this statement relates to the atropine experiment.
6. In line 98 and 183, the authors write that EM potentiates gastroparesis, but EM lowers/improves loperamide-induced gastroparesis as presented in figure 6 (GE after loperamide + EM is similar to EM, and significantly faster than after loperamide only). Please rewrite for clarity.
7. Please insert a reference in line 170 (after …previous studies), 195 (after …showed delayed GE) in line 199 (after …opioid-induced bowel syndrome).
8. Line 169-171: Please elaborate on how the results of the current study support the previous studies?
9. Line 177: remove one of the ‘that’.
10. Line 197-199 (EM therefore…(OBID)): I struggle to understand the meaning of this sentence – i.e. how does EM improve GI disorders? Can the authors please elaborate on the mechanisms/receptors involved? How does this statement relate to the results of this study?
11. Line 214: change ‘evaluated’ to ‘evaluate’.
Author Response
Response to Reviewer 1 Comments
Thank you for your helpful comments and suggestions.
Point 1: I have some issues with the statistical analyses of this study. In the study, multiple treatments (I.e. 3 or 4) are compared for the results presented in figure 3-6, however, the authors have performed pair-wise comparisons between all the treatments. To adjust for multiple comparisons, ANOVA should be performed, or, if the data are non-parametric, Kruskal-Wallis is the appropriate alternative. Only if a significant overall treatment effect is found in the analysis, the authors are allowed to perform comparison between all the individual treatments. Please refer to Festing et al, 2002, Guidelines for the design and statistical analysis of experiments using laboratory animals, ILAR journal, 43(4), 244-258. Also please describe the software and version used for analysis.
Response 1: I apologize for my mistake regarding the description of the statistical analyses. We performed ANOVA for Figures 3-6. Based on the reviewer’s suggestion, we revised the text and added “Statistical analyses were performed using the statistical software JMP® 9 (SAS Institute Inc., Cary, NC, USA). For analysis of three or more groups, analysis of variance (ANOVA) tests were performed with the Tukey post-hoc test, and analysis of differences between two normally distributed test groups was performed using Student’s t-test.” (lines 314-317). The figure legends were also changed.
Point 2: In the abstract, please describe in an introductory sentence what atropine, mosapride and loperamide are.
Response 2: Based on the reviewer’s suggestion, the sentences were revised in the abstract as follows, “Atropine (a muscarinic receptor antagonist)” (line 15), and “mosapride (a selective serotonin 5-hydroxytryptamine 4 (5-HT4) receptor agonist)” (lines 17-18), and “loperamide (a μ-opioid receptor agonist)” (lines 18-19).
Point 3: In the result section, there are a few sentences that should be moved to the introduction, i.e. line 84-86 (mosapride citrate… promote additional GE), line 92-94 (loperamide is known… induced by loperamide), and line 100-101(the levels of…. With gastric motility).
Response 3: Based on the reviewer’s suggestion, some sentences were added to the introduction as follows, “Mosapride citrate, a selective serotonin 5-hydroxytryptamine 4 (5-HT4) receptor agonist, also activates gastrointestinal tract motility; this drug is used clinically to treat gastrointestinal motility disorders. Loperamide is known to reduce gastrointestinal motility and is an effective treatment for diarrhea; this drug’s activity is mediated by the stimulation of μ-opioid receptors. We also investigated the interactions between EM and these agents.” (lines 53-58). In addition, we deleted two sentences, “Mosapride citrate, a selective serotonin 5-hydroxytryptamine 4 (5-HT4) receptor agonist, also activates gastrointestinal tract motility; this drug is used clinically to treat gastrointestinal motility disorders,” and “Loperamide is known to reduce gastrointestinal motility and is an effective treatment for diarrhea; this drug’s activity is mediated by the stimulation of μ-opioid receptors” from the results section.
Point 4: Similarly, there are a few conclusive lines in the results section, which should be moved to the discussion, i.e. line 80-82 (previous work has….action on MLNR), line 89-90 (we therefore concluded…that of mosapride), line 97-98 (we therefore concluded… loperamide-induced gastroparesis), line 111-112 (this result suggested…play complimentary roles).
Response 4: We considered moving these sentences from the results to the discussion, but we prefer to keep them in the results, because they are included in titles of results. We hope you find this acceptable.
Point 5: In the results, the authors talk about ‘higher’ gastric emptying. Would you consider changing ‘higher’ into ‘faster’, for clarity (line 68/70)?
Response 5: Yes, the word “higher” was revised to “faster” in lines 75 and 77.
Point 6: The authors conclude in line 81/82 that EM promotes GE through the vagus nerve and by direct action in MLNR, which is in agreement with the results presented in figure 3. However, this statement does not explain why EM is only able to counteract the slowing effects of atropine on GE after ip administration, and not after icv admininstation. Please elaborate how this statement relates to the atropine experiment.
Response 6: The sentence “We therefore concluded that EM promotes GE both through the vagus nerve system and by direct action on MLNR” (lines 88-89) was concluded from Fig. 4a, Fig. 4b, and reference [19]. We think that peripheral administration of the motilin receptor agonists can induce effects via any of three pathways, acting through afferent neurons, through efferent neurons, or by direct action on the muscles as described in lines 170-171 (We revised the word “mucosa” to “muscles”). However, central administration of the motilin receptor agonists can induce effects only via one pathway, through efferent neurons. Atropine blocks muscarinic receptors, which means that atropine blocks an effect via efferent neurons. So, EM is not able to counteract the slowing effects of atropine on GE after icv administration of EM. In contrast, atropine cannot counteract the direct action of EM to MLNR, so EM is able to counteract the slowing effects of atropine on GE after ip administration of EM.
Point 7:In line 98 and 183, the authors write that EM potentiates gastroparesis, but EM lowers/improves loperamide-induced gastroparesis as presented in figure 6 (GE after loperamide + EM is similar to EM, and significantly faster than after loperamide only). Please rewrite for clarity.
Response 7: The word “potentiate” in lines 101 and 187 could easily be misunderstood. The word was revised to “improve”.
Point 8: Please insert a reference in line 170 (after …previous studies), 195 (after …showed delayed GE) in line 199 (after …opioid-induced bowel syndrome).
Response 8: I inserted references in line 173 (after…previous studies), but other sentences in line 199 (after… showed delayed GE) and in line 203 (after … opioid-induced bowel syndrome) do not contain information cited from references but our results.
Point 9: Line 169-171: Please elaborate on how the results of the current study support the previous studies?
Response 9: As in our response #6, we think that central administration of the motilin receptor agonists can induce effects only via one pathway, through efferent neurons, and the effect is blocked by atropine. On the other hand, the direct action cannot be blocked by atropine in peripheral administration. These facts support the three pathways that have been suggested to exist.
Point 10: Line 177: remove one of the ‘that’.
Response 10: Thank you for pointing this out. We corrected line 181.
Point 11: Line 197-199 (EM therefore…(OBID)): I struggle to understand the meaning of this sentence – i.e. how does EM improve GI disorders? Can the authors please elaborate on the mechanisms/receptors involved? How does this statement relate to the results of this study?
Response 11: We confirmed that EM has two effects, 1) an additive effect on mosapride, and 2) it improves gastroparesis induced by loperamide. So, if symptoms of functional gastrointestinal disorders are not improved by only mosapride, EM can be considered, and if anyone suffers from OBID, EM can also be considered.
Point 12:Line 214: change ‘evaluated’ to ‘evaluate’.
Response 12: Thank you for pointing this out. We corrected line 218.

Reviewer 2 Report
This is an interesting study showing the impact of of motilin receptor agonists and ghrelin in human motilin receptor transgenic mice. The authors concluded that erythromycin and ghrelin promoted gastric emptying when administered either peripherally or centrally to transgenic mice. In general the study is well conducted and data presented support the conclusion drawn. However, a number of issues should be addressed by the authors. Issues to be addressed include (In the order they appear in the text): Materials and methods The content of fat and other details about diet composition should be provided. Please, provide details of RNA isolation and real-time PCR analyses (methods used for calculations). It is not clear how the authors performed the analyses (they only provide the primers). Do they check the feasibility of GAPDH as internal control?
Author Response
Response to Reviewer 2 Comments
Thank you for your helpful comments and suggestions.
Point 1: The content of fat and other details about diet composition should be provided.
Response 1: All mice were fed a standard rodent diet (MF, Oriental Yeast Co., Ltd., Tokyo, Japan. Energy 359 kcal, water 7.9 g, protein 23.1 g, fat 5.1 g/100 g MF). According to the reviewer’s advice, this sentence was inserted in lines 234-236.
Point 2: Please, provide details of RNA isolation and real-time PCR analyses (methods used for calculations). It is not clear how the authors performed the analyses (they only provide the primers).
Response 2: Based on the reviewer’s suggestion, some sentences were changed in the Materials and Methods as follows, “RNA was isolated using RNeasy Plus Universal Kits (QIAGEN, Hulsterweg, Netherlands). RNAs were converted into cDNAs using a High-Capacity cDNA Reverse Transcription Kit (Thermo Fisher Scientific, Waltham, MA, USA). cDNA products were quantified by real-time PCR using Light Cycler SYBR Green (Roche, Basel, Switzerland) and a Light Cycler®480 (Roche, Basel, Switzerland), with the gene encoding glyceraldehydes-3-phosphate dehydrogenase (GAPDH) as an internal control.” (lines 250-254).
Point 3: Do they check the feasibility of GAPDH as internal control?
Response 3: We previously checked the feasibility of GAPDH [Takeshita, E. et al. J Gastroenterol. 2006;41:223-230.] [Nunoi, H. et al. Regul Pept. 2012;176:28-35.]. GAPDH is often used as an internal control for intestine [Bing, Z. et al. Nat Commun. 2019;10:1071] and brain [Xiurong, Z. et al. Nat Commun. 2017;8:602].

Reviewer 3 Report
Re ijms-451166
Effects of motilin receptor agonists and ghrelin in human motilin receptor transgenic mice.
Mice are generally motilin system knockout animals lacking receptor and motilin different from the case of human. Therefore, mice are not used for an animal model for examination of the effect of motilin and interaction of motilin with ghrelin. Therefore, Suncus is proposed to be an amimal model expressed both motilin and ghrelin system, and interaction of both peptides were reported. Kawamura et al produced the human motilin receptor transgenic mice for investigation of the effect of motilin receptor agonist, EM in vivo. Transgenic mice responded to EM and increased the gastric emptying underlying mechanisms slightly different between ICV and IP injection.
This paper is very simply designed and the data clearly showed the results. The transgenic mice is a useful animal model to investigate the action of motilin agonists in mice. My comments to be addressed by the authors are followings:
1) Results of the paper must be interesting to understand the prokinetic action of motilin agonist, but the condition of mice is not physiological because there are no endogenous motilin despite the expression of motilin receptor. It is not possible to examine the effect of EM on endogenous motilin release. How do you check this possibility in other animals? Are there any influence on motilin receptor function under long time lacking of receptor ligand in the mice?
2) Is there any interaction between ghrelin and EM in the gastric emptying? It is not described in the manuscript. Positive functional interaction is demonstrated in Suncus but negative interaction is in dogs. Please add the data.
3) Atropine did not decrease the response of EM applied ip. Does it mean the direct action of EM on the smooth muscle cells? Have you checked it using in vitro strips?
4) Please add the data that ip (peripherally) injection of EM changes the plasma ghrelin level or not. If the central application of EM only effective to decrease the plasma ghrelin level, what is mechanisms of this decrease? EM looks like to decrease the expression of ghrelin mRNA. Does atropine recover the decrease of ghrelin like an inhibition of EM-induced action on gastric emptying? You described that EM did not affect the gastric emptying in WT mice. Is it the same in the decrease in the ghrelin? Please discuss the detailed mechanisms of decrease in the ghrelin contents in stomach and plasma.
Author Response
Response to Reviewer 3 Comments
Thank you for your helpful comments and suggestions.
Point 1: Results of the paper must be interesting to understand the prokinetic action of motilin agonist, but the condition of mice is not physiological because there are no endogenous motilin despite the expression of motilin receptor. It is not possible to examine the effect of EM on endogenous motilin release. How do you check this possibility in other animals? Are there any influence on motilin receptor function under long time lacking of receptor ligand in the mice?
Response 1: We showed that EM acts directly on MLNR and gastrointestinal muscles were contracted [Kato, S. et al. PLoS One. 2019;14:2:e0205939.], and motilin does not induce endogenous release by EM. We previously confirmed that EM acts by binding to transmembrane helical domains of the human MLNR [Utsunomiya, S. Matsuura, B. et al. Regul Pept. 2013;180:17-25.]. We also previously confirmed that Tg mice showed the same phenotype as WT mice, (e.g. amount of food, water, BW) in the absence of ligands [Nunoi, H. Matsuura, B. et al. Regul Pept. 2012;176:28-35.]. Moreover, WT mice showed no responses, although we attempted administration of EM and motilin. We do not think there is any influence on motilin receptor function in mice lacking the receptor for a long time.
Point 2: Is there any interaction between ghrelin and EM in the gastric emptying? It is not described in the manuscript. Positive functional interaction is demonstrated in Suncus but negative interaction is in dogs. Please add the data.
Response 2: We don’t’ have the data you instruct. EM improves gastric emptying as same as motilin. We referenced past reports about Suncus [Kuroda, K. et al. Endocrinology. 2015;156:4437-4447.][Takemi, S. et al. Endocr J. 2017;64:S11-14.] in the manuscript. (Line204, references[37][38]) In humans, the peak of plasma ghrelin appears in fasting [Shiiya, T. et al. JCEM 87:240:2002], in contrast, the peak of plasma motilin appears 2 hours after eating [Boivin, M. et al. J Gastroint Motility. 2;240,1990]. Our data showed the complementary roles of EM and ghrelin in centrally EM-administrating in Tg mice. We want to try your idea in the near future, thank you very much.
Point 3: Atropine did not decrease the response of EM applied ip. Does it mean the direct action of EM on the smooth muscle cells? Have you checked it using in vitro strips?
Response 3: Yes, we have tested this in muscle strips [Kato, S. et al. PLoS One. 2019;14:2:e0205939].
Point 4: Please add the data that ip (peripherally) injection of EM changes the plasma ghrelin level or not. If the central application of EM only effective to decrease the plasma ghrelin level, what is mechanisms of this decrease? EM looks like to decrease the expression of ghrelin mRNA. Does atropine recover the decrease of ghrelin like an inhibition of EM-induced action on gastric emptying? You described that EM did not affect the gastric emptying in WT mice. Is it the same in the decrease in the ghrelin? Please discuss the detailed mechanisms of decrease in the ghrelin contents in stomach and plasma.
Response 4: The data of plasma ghrelin level in Tg mice peripherally administered EM showed large variation. So, we showed only the data of centrally administered EM. The mechanism of the decrease in ghrelin level and ghrelin-positive cells due to centrally administered EM is unclear, as described in the discussion. We want to elucidate the phenomenon in the future.
We do not mention the expression of ghrelin mRNA in the manuscript. We examined it, but there was no significant difference.
We have not examined the level of acyl-ghrelin using atropine in Tg mice, or the effect of EM on the levels of acyl-ghrelin in WT mice. We think that elucidating the mechanism is very important, although it is currently unknown, as described in discussion.

Round 2
Reviewer 1 Report
Thanks to the authors for their revisions.
Author Response
Thank you for your considerations.
Reviewer 2 Report
The authors addressed the concerns of this RW
Author Response
Thank you for your considerations.
Reviewer 3 Report
Title, Effects of motilin receptor agonists and ghrelin in human motilin receptor transgenic mice, is not clear what the authors observed in this experiment. Please insert that the authors examine the effect of EM and ghrelin on gastric emptying of human motilin receptor transgenic mice. As I mentioned before, it is better to describe the gastric strips of trangenic mice can respond to EM and motilin but not ghrelin. These in vitro data are useful to understand this in vivo data.
Author Response
Response to Reviewer 3 Comments
Thank you for your helpful comments and suggestions.
Point 1: Title, Effects of motilin receptor agonists and ghrelin in human motilin receptor transgenic mice, is not clear what the authors observed in this experiment. Please insert that the authors examine the effect of EM and ghrelin on gastric emptying of human motilin receptor transgenic mice.
Response 1:
Thank you for your advice. We considered the title should be changed or not. However, we prefer to keep the current title, because we investigated not only effects of gastric emptying by motilides and ghrelin but also relation between EM and ghrelin. We hope you find this acceptable.
Point 2: As I mentioned before, it is better to describe the gastric strips of transgenic mice can respond to EM and motilin but not ghrelin. These in vitro data are useful to understand this in vivo data.
Response 2:
We have not tested the response of gastric strips of Tg mice by ghrelin stimulation. We have already mentioned that EM and motilin have a direct action to MLNR in isolated stomach (the gastric strips) in the discussion (line 224-227) [Kato, S. et al. PLoS One. 2019;14:2:e0205939], and have mentioned that EM’s effect on the contractile response in the (isolated) stomach of Tg mice reflects direct action via MLNR (line 87-88) [Kato, S. et al. PLoS One. 2019;14:2:e0205939].
We revised the reference number in line 173 from “16,26-28” to “19,26-28”.
